# Tetraplex Fluorescent Microbead-Based Immunoassay for the Serodiagnosis of Newcastle Disease Virus and Avian Influenza Viruses in Poultry Sera

**DOI:** 10.3390/pathogens11091059

**Published:** 2022-09-17

**Authors:** Na Zhao, Christian Grund, Martin Beer, Gang Wang, Timm C. Harder

**Affiliations:** 1Institute of Diagnostic Virology, Friedrich-Loeffler-Institute, Suedufer 10, 17493 Greifswald, Germany; 2Precise Genome Engineering Center, School of Life Sciences, Guangzhou University, Guangzhou 510030, China

**Keywords:** Newcastle disease virus, avian influenza, serosurveillance, fluorescent microspheres, immunoassay, multiplex detection

## Abstract

Virulent Newcastle disease virus (NDV) as well as highly pathogenic avian influenza (HPAIV) subtypes H5 and H7 induce contagious and lethal systemic disease in poultry. In contrast, low pathogenic AIV H5 and H7 may circulate clinically unnoticed in poultry but eventually generate HPAIV. Low pathogenic NDV strains are widely used as live-attenuated vaccines against ND. Serological tools are essential to conduct active surveillance for infections with notifiable AIV-H5, -H7 and to control vaccination against NDV and HPAIV in poultry populations. Here, recombinant nucleocapsid proteins (NP) of AIV and NDV, and haemagglutinin protein fragment-1 (HA1) of AIV subtypes H5 and H7 were expressed in *E. coli*. Purification and refolding were required before coating fluorescent microspheres via streptavidin-biotin linkage. The tetraplexed inhibition fluorescent microsphere immunoassay (iFMIA) was then assembled for analysis on a Luminex^®^-like platform (Bioplex^®^) using murine monoclonal antibodies specific for each of the four targets. The assay was evaluated by testing galliform poultry sera derived from experimental infections (n = 257) and from farms (n = 250), respectively. The tetraplex iFMIA compared favorably with commercially available ELISAs and the “gold standard” hemagglutination inhibition assay. Tetraplexed iFMIA provided a specific and sensitive tool to detect and discriminate AIV- and NDV-specific antibodies in the sera of galliform poultry.

## 1. Introduction

Newcastle disease in poultry is caused by virulent forms of the avian orthoavulavirus-1 (AOAV-1), commonly referred to as Newcastle disease virus (NDV) [1]. Avian influenza viruses (AIV) are influenza type A viruses which hold generic status in the Orthomyxoviridae family. So far, sixteen subtypes of the hemagglutinin (HA) and nine of the neuraminidase (NA) proteins of AIV have been distinguished in avian species [2]. Velogenic/virulent Newcastle Disease (ND) caused by infections with NDV and highly pathogenic influenza A (HPAI) of subtypes H5 or H7, respectively, range among the most-dreaded poultry diseases worldwide. Consequently, infections in poultry with AIV subtypes H5 and H7 of both low and high pathogenicity, and with velogenic ND, are globally notifiable to the World Organization of Animal Health (WOAH). The tight control of these infections requires (i) the prevention of virus incursions by advanced biosecurity measures; (ii) the detection of outbreaks as early as possible using molecular diagnostics; (iii) the suppression of further virus spread by harsh restriction measures; (iv) elimination from affected poultry populations by, ultimately, culling affected holdings and vaccination [3]. However, these viruses remain enzootic in several regions of the world and continue to threaten industrial as well as backyard poultry rearing on a broad scale [2,3].

Vaccination with inactivated or live virus vaccines featuring low virulent (lentogenic) NDV strains such as LaSota or B1 is a widely used preventive measure against ND [4]. In some countries, including Germany, the vaccination of all gallinaceous poultry against NDV is mandatory. Vaccination against notifiable AI (subtypes H5 and H7) employing inactivated adjuvanted vaccines or recombinant modified-live virus vaccines, including NDV as a backbone [5] and expressing the HA of AIV, is used more restrictedly [3].

The serological monitoring of poultry populations for NDV and AIV subtypes H5/H7 is an important tool to estimate infection prevalence or to evaluate the efficacy of vaccination campaigns [6,7,8]. A number of serological assay formats such as agar gel immunodiffusion (AGID), hemagglutination inhibition (HI), and enzyme-linked immunosorbent assays (ELISA) are available for this purpose [9,10,11]. The AGID test detects precipitating antibodies and, although it is a robust test format, it suffers from limited sensitivity and cannot be used with the sera of waterfowl [10]. The HI assay detects antibodies that specifically inhibit the hemagglutination of NDV and AIV H5/H7 due to interference with the receptor-binding of the NDV hemagglutinin-neuraminidase (HN) or the AIV HA proteins, respectively; this assay remains the “gold standard” for detecting NDV or influenza virus subtype-specific antibodies [12,13]. Unfortunately, HI assays are labor intensive, require the production of antigens from potentially hazardous viruses, and, due to the high specificity of the assay, sensitivity may suffer if the antigenic profiles of the test antigen and the field viruses in circulation do not sufficiently match [8,14]. ELISA-based assays for detection antibodies may overcome many of these limitations: They are easy and safe to use and can be tailored more flexibly to the diagnostic needs. Several ELISA kits that are specific for the NDV nucleocapsid protein (NP) or the AIV NP, H5 and H7 HA proteins are commercially available [6,9,15]. However, such solid phase ELISAs are limited to detecting antibodies against a single target and demand substantial costs (and time) when antibodies against all four of the targets mentioned above are to be measured.

The Luminex^®^ suspension array system utilizes xMAP^®^ technology and permits the multiplex probing of up to five hundred different analytes within a single reaction. Luminex technology enables the fast, accurate, and flexible measurement of multiple biomarkers such as RNA or protein targets simultaneously in a single sample [16,17,18,19,20,21,22]. This technique is based on distinct fluorescent color-coded beads that can be conjugated with analyte-specific targets. Multiplexing is achieved by mixing different distinctly coded analyte-specific beads that are probed in a liquid suspension assay. This assay format combines the advantages of solid phase ELISAs with the benefits of a lower sample volume, reduced costs and turn-around time if multiplexing is applied.

Here, we explored a tetraplex inhibition fluorescent microsphere immunoassay (4plex iFMIA) based on Luminex^®^ bead technology to simultaneously detect the NP-specific antibodies of NDV and AIV, as well as AIV H5- and H7-specific antibodies using recombinantly expressed proteins as targets in a blocking liquid array format.

## 2. Materials and Methods

### 2.1. Animal Experimentation Permits

All animal experiments for the production of reference sera received full legal approval by the animal welfare committee of the German Federal State of Mecklenburg-Western Pomerania (LALLF M-V/TSD/7221.3-2.5-004/10; LALLF M-V/TSD/7221.3-2.5-010/10).

### 2.2. Virus Propagation

NDV vaccine strain LaSota (LS) and AI viruses were grown in the allantoic cavity of embryonated chicken eggs, as detailed elsewhere [15,23]. Recombinant influenza A virus nucleoprotein was produced from A/swine/Germany/R1738/2010 (H1N1, [EPI426141]) and recombinant NDV nucleoprotein was derived from NDV vaccine LaSota. Truncated AIV recombinant hemagglutinin proteins originated from A/turkey/Germany/R1612/08 (H5N3 LPAI) and A/chicken/Germany/R28/03 (H7N7 HPAI, [AJ620350]).

### 2.3. Cloning and Bacterial Expression of Recombinant Proteins

Recombinant pET19b vector was used to express ND and AI proteins in Rosetta-gami *E. coli* cells (Novagen, Darmstadt, Germany). Full length ORFs encoding the NP of NDV LaSota (489 amino acids) and AIV (498 amino acids) were cloned into pET19b downstream of the T7 promoter using primer-directed techniques. An octa-histidine and an Avi-tag, the latter used as a biotin acceptor site [24], were positioned N-terminally of the NP (Figure 1). Similarly, fragments of the hemagglutinin HA1 coding part representing amino acids (aa) 17–342 (H5) and 19–337 (H7), respectively, of the above mentioned AIV isolates were cloned into pET19b (Figure 1). Plasmid pBirCam (Avidity, Aurora, CO, USA) over-expressing the bacterial biotin ligase BirA was co-transformed with either of the four constructs into *E. coli* strain Rosetta-gami to ensure the co-translational mono-biotinylation of the recombinant NP and HA1 proteins at the lysine residue of their respective Avi-tag [6,7]. Ampicillin and chloramphenicol were used for selection, and plasmid insert-specific PCRs were used for the identification of double transformants.

The induction of expression was achieved in tryptone-yeast-hepes (TYH) medium (20 g of tryptone; 10 g of yeast extract; 5 g of NaCl; 1 g of MgSO_4_; 11 g of HEPES in 1 L aq. bidest.) supplemented with IPTG (1.5 mM) and D (+)-biotin (50 µM). After culturing for 4 h at 37 °C, the cells were pelleted and lysated using ultrasonic disruption, as previously described [25], in order to liberate the inclusion bodies (ICs) into which the recombinant proteins had sequestered.

### 2.4. Purification and Reconstitution of Recombinant Proteins from Bacterial Inclusion Bodies

Purified inclusion bodies (ICs) were solubilized in 6 M of guanidinium-HCl. A set of refolding buffers was used in a stepwise solubilization strategy (ProteoStat kit, Enzo, Lörrach, Germany). Finally, an optimized refolding buffer was identified consisting of 50 mM of Tris, a pH of 8.3, 20 mM of NaCl, 0.8 mM of KCl, 0.8 M of L-Arginine, and 0.12 M of sucrose. Antigenic properties in solid phase ELISA were assayed using polyclonal antibodies raised in chickens or with specific monoclonal antibodies. The final protein concentration in refolding buffer was determined using a Coomassie protein assay kit (ThermoScientific, Rockford, IL, USA) and proteins were stored at 4 °C until further use.

### 2.5. Production of Positive Control Sera

Serum S185, a hyperimmune serum raised in chicken against inactivated NDV strain Ulster, served as a standard positive control for assays employing NDV rNP. Likewise, hyperimmune serum S304, produced against the AIV isolate A/duck/Italy/636/2003 (H7N3) and serum S82, raised against the HA of A/Vietnam/1194/2004 H5N1 (expressed by recombinant A/Puerto Rico clone NIBRG-14; kindly provided by Dr. J. McCauley, Mill Hill, UK, in the frame of the WHO PIP program) were used for assays employing AIV rNP, H5-rHA1, or H7-rHA1. An SPF chicken serum was chosen as a negative control. The immune status of these control sera was assessed by three different methods (Appendix A). These control sera were used as standards in each of the assays to determine validity (threshold Fl (fluorescence intensity)) and to calculate S/N ratios.

### 2.6. Production of Experimental Infection or Vaccination Sera in Chickens or Turkeys

Table 1 gives an overview of the status of different sera series used for the evaluation of the 4plex iFMIA assay. The sera were produced in chickens in the frame of several animal experiments conducted at FLI. A total of 257 sera, which included 80 SPF chicken sera, was used.

### 2.7. Origin of Field Sera Not Used for Reference Purposes

A total of 150 chicken and 130 turkey sera submitted for routine AI or ND serodiagnosis originated from various poultry holdings in Germany and were investigated by 4plex iFMIA assay and other routine assays in this study.

### 2.8. Monoclonal Antibodies (mAbs)

A mAb-horse radish peroxidase (HRPO) conjugate specific for biotin was purchased from NEB (#7075). MAb NP-36, specific for the NP protein of NDV, had been generated against NDV strain LaSota and was kindly provided by Dr. B. Koellner, Friedrich-Loeffler-Institute, Riems, Germany [26]. The NP protein of influenza A viruses is recognized by mAb HB-65, which was purchased from ATCC (H16-L10-4R5). The AIV HA H5 specific monoclonal antibody 8292, generated against recA/Vietnam/1194/2004 (NIBRG-14), was provided by Dr. R. Ehricht (Alere, Jena, Germany), while Mab 15C7, raised against an unknown low pathogenic AIV isolate of subtype H7, was kindly supplied by Dr. P. Pourquier (ID-Vet, Montpellier, France) [27]. A phycoerythrin (PE)-labeled mAb directed against hexahistidine was purchased from Abcam (Ad1.1.10; Cambridge, UK).

### 2.9. Western Blotting

The recombinant proteins were separated by denaturing SDS-PAGE in 12.5% gels, then transferred onto nitrocellulose membranes by semi-dry blotting at 0.8 mA/cm^2^ for 1 h (PerfectBlueTM Electro Blotter, Peqlab, Erlangen, Germany). Membranes were blocked with 5% skim milk powder in Tris-buffered saline (TBS) supplemented with 0.05% (*v*/*v*) Tween 20 (TBST) for 1 h. Specific mAbs or sera were appropriately diluted in TBST and incubated on the membranes for one hour at room temperature. The membranes were washed three times with TBST and incubated for another hour with anti-murine IgG or gallid IgY HRPO-labeled secondary antibody conjugate (Santa Cruz Biotechnology, Heidelberg, Germany). Blots were washed three times and incubated with SuperSignal™ West Pico Chemiluminescent substrate solution (ThermoScientific, Braunschweig, Germany) before being analyzed on an imaging system (VersaDoc, Bio-Rad, Munich, Germany). Photos were edited with respect to contrast and brightness and composite blots were assembled using cut-out lanes (GIMP software, version 2.10.32).

### 2.10. Coupling of Recombinant Antigens to Streptavidin Precoated Luminex Beads

Refolded, biotinylated recombinant proteins were coupled to individual streptavidin precoated Lumilite^®^ beads according to the manufacturer’s instructions (Progen Biotechnik Corporation, Heidelberg, Germany). Briefly, the recombinant proteins NDV-rNP, AIV-rNP, AIV subtype H5-rHA1 and AIV subtype H7-rHA1 were coupled with 2 × 10^6^ fluorescent beads (Lumilite^®^ MMLL03-10, -23, -50, -56) by 100 µg, 200 µg, 25 µg and 20 µg of each antigen, respectively, shaking for more than 2 h at room temperature in the dark. Then, the beads were washed three times in assay buffer (PBS (pH 7.4) supplemented with 1% (*w*/*v*) biotin-free bovine serum albumin and 0.05% (*v*/*v*) Tween 20). After coupling, the antigen-coated beads were stored in the dark at 4 °C. Microspheres were protected from the light throughout the whole procedure.

### 2.11. Multiplex Bead-Based Immunoassay

Multiscreen HTS IP 0,45 µm 96-well filter plates (Millipore, Bedford, MA, USA) were prewashed once with 150 μL of assay buffer, and washing fluids were aspirated using a vacuum manifold (Millipore, Bedford, MA, USA). Recombinant antigens coated beads were sonicated and vortexed for 20 s to disrupt the aggregation, then mixed and diluted to a final concentration of 4 × 10^4^/mL beads each in assay buffer. Finally, 50 µL of this suspension containing 2000 beads/antigen was used per well of the filter plate in this assay. Next, serum samples diluted 1:20 in 50 µL of assay buffer were applied to the plates and incubated for 40 min. Then, the solution was removed and the beads were washed. Meanwhile, a set of four previously selected reference sera (S185, S82, S304 and SPF, (Appendix A)) were run on each plate as inter-assay and controls. After that, the selected monoclonal antibodies against four antigens diluted properly individually were mixed together, and 50 µL was taken into each well for incubation for 30 min. After washing, 50 μL of diluted 1:4000 PE-labelled polyclonal antibodies raised in goats against murine IgG (Fab’)2 (Santa Cruz Biotechnology, Heidelberg, Germany) was added to each well and incubated for 30 min. After final washing, the beads were resuspended in 125 μL of assay buffer, and plates were placed on the shaker for 15 min to counteract beads aggregation. In this assay, the washing step was performed three times with 200 μL of assay buffer, and the solution was aspirated from the plate by the vacuum manifold. All the incubation steps were carried out on a rocking platform in the dark at room temperature. The samples were measured in a BioPlex 100 instrument (Bio-Rad, Munich, Germany) running the Bioplex Manager 6.1 software (Bio-Rad, Munich, Germany). The instrument was calibrated using the Bio-Rad calibration sets CL1/CL2 before each use, and 50–100 beads per species and samples were measured in a sample volume of 50 μL. Throughout, the results were calculated and presented as S/N ratios, in which the S value arises from the median fluorescence intensity (MFI) of the samples, and N from the MFI of the negative control.

### 2.12. Commercial ELISAs for Detection of AIV and NDV Antibodies

A commercial indirect NDV-ELISA (Flock Chek NDV) based on complete ND virions was purchased from IDEXX (Ludwigsburg, Germany). The manufacturer’s instructions were followed exactly using either the version for chickens or turkeys as appropriate. Commercial AIV competitive ELISA kits (ID.Vet, Montpellier, France) were used for AIV NP, H5 or H7 specific antibody detection in poultry sera. Assays were performed according to the manufacturer’s protocols. For the NDV assay, the sera were considered positive if the S/P ([sample-negative control]/[positive control-negative control] OD450) ratio was >0.2. For the AIV assay, the inhibition values were calculated by [Sample OD450/negative control OD450] × 100, then compared with the recommended cut-offs by the manufacturer to determine the sample sera as “positive”, “questionable” or “negative”.

### 2.13. Hemagglutination Inhibition Assay (HI)

HI assays were performed according to O.I.E. recommendations, as described by Grund [9]. Briefly, four hemagglutinating units of the inactivated antigens of NDV strain LaSota or AIV strains A/common teal/England/7894/2006 (H5N3) and A/turkey/England/647/1977 (H7N7) were used. The AIV antigens were produced at the Community Reference Laboratory for Avian Influenza at APHA, Weybridge, UK, and are used in the European sero-monitoring program for poultry. All sera had been heat-inactivated for 30 min at 56 °C. Serum HI titers > 1:8 were considered positive.

### 2.14. Statistical Analysis

For the four serological assay systems held here as a gold standard (HI NDV, HI H5, HI H7, ELISA NDV), the prevalence was estimated separately for the groups of sera from experimental infections (EXP) or from the field (F). Cut-off values for the 4plex iFMIA were selected according to receiver-operating-curve analyses (ROC) with regard to the criterion “minimum ROC distance”, combining groups EXP and F. Sensitivity and specificity were calculated on the basis of the gold standard assays, and for each system, all the assays were compared pairwise using kappa statistics; EXP and F groups were treated separately. Calculations were performed using the program “R” (version 2.13.0) with the package Diagnosis Med, Version 0.2.3.), according to the R Development Core Team [28].

## 3. Results

### 3.1. Bacterial Expression of Recombinant Proteins

Recombinant proteins were purified from bacterial inclusion bodies, denatured and refolded in vitro. Biotinylated protein products matching the predicted molecular weights of 56.1 kDa (NDV-rNP), and 59.2 kDa, 39.7 kDa, and 37.9 kDa for AIV-rNP, rH5-HA1, and rH7-HA1, respectively, were detected by Western blotting using a biotin-specific mAb (Figure 2). The proteins also reacted specifically with positive polyclonal chicken control sera against NDV and AIV H5/H7 viruses (Figure 2B–D), although these were raised against heterologous antigens. The influenza rNP antigen, although derived from a porcine H1N1 virus, was recognized by sera raised against avian-origin influenza A viruses of subtypes H5N3 and H7N7 (lanes 1 of Figure 2C,D). The NDV-rNP antigen originating from the LaSota vaccine strain reacted with sera produced against inactivated NDV strain Ulster (lane 4, Figure 2B). Similarly, a chicken serum raised against H5N1 was specific for the rHA1 protein of H5N3 (lane 2, Figure 2D) and serum S304 raised against H7N3 was specific for the rHA1 proteins of H7N7 (lane 3, Figure 2). The results indicate that the recombinant proteins of the expected sizes and broadly specific antigenicity were successfully expressed, purified and refolded.

To establish the inhibition fluorescent microsphere immunoassay, the specific epitopes within these four antigens, which can be recognized by monoclonal antibodies, should be preserved properly after refolding. A solid phase indirect ELISA format was used to verify authentically refolded epitopes. Briefly, the four recombinant proteins were coated in a 96-well plate each. Then, the dilution series of the four mAbs was added, and finally, a HRPO-conjugated antibody against mAbs was detected to evaluate the interactions between recombinant proteins and their corresponding mAb; TMB substrate was used. As a result, all four recombinant antigens were recognized at high dilutions by the specific mAbs (data not shown).

### 3.2. Development of the 4Plex iFMIA Assay

The principle of our 4plex iFMIA assay used in this study is depicted in Figure 3. The biotin labeled recombinant proteins were coupled to streptavidin-precoated Luminex beads to set up the 4plex iFMA assay. To optimize the quantity of antigens coupled, a series of different concentrations of recombinant proteins coated with beads were assayed by a PE-labeled mAb against HIS tag, which were measured in the BioPlex 100 instrument. The quantity of protein was considered optimal if the MFI values were between 2000–6000 (data not shown). As a result, the optimal quantity of NDV-rNP, AIV-rNP, AIV H5-rHA1 and AIV H5-rHA1 was 50 µg, 100 µg, 12.5 µg and 10 µg of protein/106 beads, respectively. These protein quantities were used for all subsequent experiments. Next, the checker-board titration test was applied to optimize further assay conditions, i.e., the dilutions of the serum sample, each mAb (against four antigens) and PE-labelled polyclonal anti-murine antibodies. Finally, the sample sera were diluted 1:20 and the second PE-labeled antibody 1:4000; the four mAbs against NDV-rNP, AIV-rNP, AIV H5-rHA1, AIV H5-rHA1 were used at dilutions of 1:40, 1:320, 1:3000, and 1:4000, respectively. Finally, the Luminex beads were resuspended in the assay buffer and analyzed in the Bioplex 100 instrument, which enabled us to identify the unique fluorophore-encoded beads and the PE fluorescence signal of each bead. Due to the mAb-directed blocking principle described above, this assay is species independent.

### 3.3. Four Plex iFMIA Cutoff Determination and Performances Compared with ELISAs and HI

To set the cutoff values for the 4plex iFMIA, a series of sera derived from experimental NDV/AIV infected and/or vaccinated animals with different immune status were tested (Table 1). An ROC curve analysis was used to analyze the data and calculate cutoff values for each of the four antigens, in which the HI assay was used as the “golden standard”. As a result, the cutoff values of NDV-rNP, AIV-rNP, H5-rHA1, and H7-rHA1 were 91.8, 75.4, 61.3, and 39.4, respectively (Appendix A).

According to the cutoff values above, the sensitivity and specificity for the 4plex iFMIA assay for recombinant proteins NDV-rNP, AIV-rNP, H5-rHA1, and H7-rHA1 were calculated individually under a 95% C-Index (CI) condition by ROC curve analyses. The results are presented in Table 2. The AUCs (area under the ROC curve; values equal to 1 represent perfect performance) measured 0.937 for NDV-rNP, 0.987 for AIV-rNP, 0.996 for H5-rHA1, and 0.986 for H7-rHA1. These data indicate that the 4plex iFMIA assay exhibited promising sensitivity and specificity using sera derived from experimentally infected poultry. Meanwhile, commercial ELISAs for each pathogen were also used to detect the specific antibodies of these experimental sera. Thereafter, the data were also analyzed by an ROC similar to those of iFMIA for comparison purposes (Table 2). The results reveal that for all four targets, the performance characteristics of the 4plex iFMIA compared favorably with the commercial ELISAs when using HI titers as a gold standard. It should be noted that for the detection of H5- and H7-specific antibodies, the 4plex iFMIA showed a slightly increased sensitivity compared to commercial ELISA. According to these cutoff values, we re-determined the number of positive samples among the sera obtained from experimental infections/vaccinations. The results from this analysis are summarized in Table 3. Here, in seven groups of sera with different immune status, the rates of seropositives as analyzed by HI, ELISA, and iFMIA assay revealed a high level of congruence.

Furthermore, the sera from thirty SPF chickens that received immunization with a live-attenuated vaccine based on NDV strain “Clone 30” were examined for the kinetics of antibody development (Figure 4). The results indicate that the NDV NP specific seroresponse can be accurately detected by the 4plex iFMIA format, similar to commercial ELISA and homologous HI assay. Simultaneously, the chickens remained seronegative for AIV by 4plex iFMIA and the other assays, indicating a high level of congruence among the three assay formats (data not shown).

### 3.4. Performance Characteristics of the 4Plex iFMIA Assay with Field Sera

A total of 250 field sera collected from commercial chickens and turkeys in various regions of Germany was tested by HI, commercial ELISAs, and 4plex iFMIA assays. The results are presented in Table 4. Antibody detection by iFMIA for AIV H5 and H7 in field serum samples exhibited a high sensitivity and specificity (0.99 and 1), similar to results obtained with the sera of experimental infections. For AIV-rNP, the sensitivity of the 4plex iFMIA format declined slightly from 0.97 when testing experimental sera, to 0.87 when testing field sera. For the NDV-rNP specific iFMIA, the number of field sera scoring positive was excessively higher compared to HI, resulting in a decrease in specificity from 0.91 to 0.56. A similar situation was observed when using the commercial ND ELISA assay and comparing the results to the HI assay (Table 4).

## 4. Discussion

In this report, we demonstrate the feasibility of developing a Luminex microbead-based liquid immunoassay for simultaneously detecting specific antibodies against avian pathogens NDV and AIV of subtypes H5 and H7 in a single serum sample. The 4plex iFMIA represents an interesting alternative to the aforementioned assays. The time required to measure 96-well plates and the dependency on Luminex^®^/Bioplex^®^ machinery are the current constraints of the 4plex iFMIA. In general, the economic efficacy of multiplexed iFMIA will increase with the number of sera to be examined at the same time, as a calibration of the Bioplex^®^ instrument is an essential time-consuming prerequisite for each run. An optimization of the number of beads and flow of beads to be measured is pivotal in economizing time for scanning. Moreover, care must be taken to avoid aggregating the beads. As such, work is still required to render 4plex iFMIA feasible for commercial high-throughput surveillance testing.

In contrast to the results obtained with ND positive sera from experimentally vaccinated birds, there was less congruence of the ND-specific results for avian field sera with the results of the HI assay, resulting in lower specificity. It should be noted that, in Germany, NDV vaccination is compulsory for all gallinaceous poultry, including those kept in small backyard herds; all field sera originated from flocks in Germany. A similar effect was observed using the commercial ND ELISA when compared with HI. This discrepancy has also been observed by previous studies [15,30]. Although the HI assay is considered a “golden standard”, it measures antibodies that inhibit hemagglutination by the HN ND viral protein, whereas both the 4plex iFMA format and the commercial ELISA measured NP-specific antibodies. It may be speculated that ND vaccination in the field may induce a higher level of NP- versus HN-specific antibodies, and/or that NP-specific antibodies have slower declining kinetics post-vaccination [15,30,31]. In any case, the HI assay did not prove an ideal control in the ROC analysis in the field sera measurements.

To our knowledge, only a few further reports on the use of fluorescent microsphere technology for the detection of ND- and AI-specific antibodies in avian sera have been published [32,33,34,35]. These assays were limited to the duplex detection of antibodies directed against the ND and AI NP [32,35], or AI NP and H5 [34], or the triplex detection of AI NP, M1, and NS proteins [33]. Recombinant antigens expressed by either the baculovirus or an alphavirus replicon system were used in those studies. Performance characteristics comparable to the values obtained in this study were reported. The currently developed 4plex iFMIA system provides advantages over the published systems: (i) The use of bacterially expressed recombinant antigens that are mono-biotinylated in vivo allows the production of larger amounts of target proteins in less demanding bacterial culture systems. (ii) An inhibition format is employed that is based on the use of specific mAbs for each of the targets. This, in turn, will allow the testing of avian sera independent of the poultry species. Previous assays depended on species-specific conjugates. (iii) The simultaneous detection of generic and subtype-specific antibodies against notifiable AI is facilitated.

Demands to examine an ever-growing number of infection-related serological analytes in small sample volumes have constantly increased. With respect to the notifiable avian viral diseases AI and ND, at least four serological parameters need to be measured in surveillance programs to study disease prevalence or to control vaccination efficacy at the population level. Attempts to multiplex such assays are a logical consequence. Fluorescent microsphere systems offer vast opportunities in this respect and should be more intensely explored in the future. The current work could be extended to include the possible quantification of viral/antigen loads, and the detection of antibodies against other AIV subtypes (e.g., the H9), further NDV proteins (e.g., the HN), and other significant avian virus diseases (e.g., the infectious bronchitis or infectious bursal disease viruses).

## Figures and Tables

**Figure 1 pathogens-11-01059-f001:**
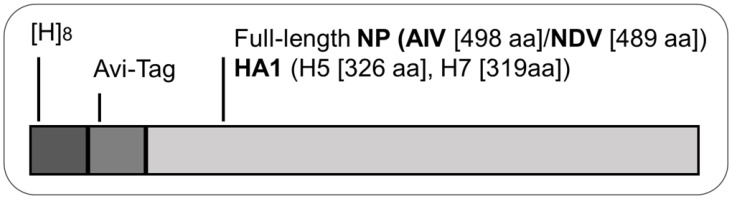
Schematic presentation of constructs used for the generation of pET19b plasmids and T7-driven expression of NDV and AIV proteins in Rosetta-gami *E. coli* cultures. Protein N-terminus is shown on the left with N-terminal octahistidin ([H]8) and Avi-tag (15 aa). The Avi-tag is used for mono-biotinylation in vivo. “aa”—amino acid.

**Figure 2 pathogens-11-01059-f002:**
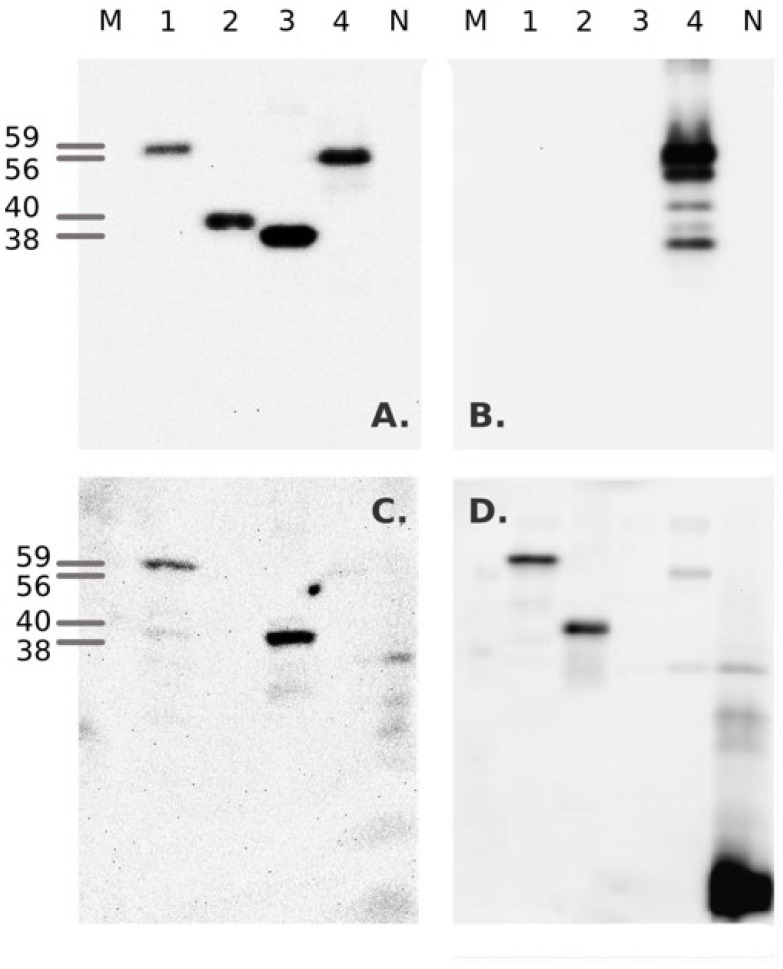
Assembled Western blots of recombinant, in vivo biotinylated HA1 and NP proteins of avian influenza and Newcastle Disease viruses. M—protein size standards (the size of relevant proteins is indicated to the left), 1—rNP_R1738/10; 2—rHA1_H5_R1612/11; 3—rHA1_H7_R28/01; 4—rNP_NDV_LS. N—lysate of *E. coli* Rosetta-gami cells transfected with an empty pET19b vector and induced with IPTG. Blots were stained as follows: (**A**)—anti-biotin monoclonal antibody; (**B**)—chicken serum S185 raised against NDV strain Ulster; (**C**)—chicken serum S304 produced against A/duck/Italy/636/2003 H7N3; (**D**)—chicken serum S82 against rHA A/Vietnam/1194/2004 H5N1 [NIBRG-14]).

**Figure 3 pathogens-11-01059-f003:**
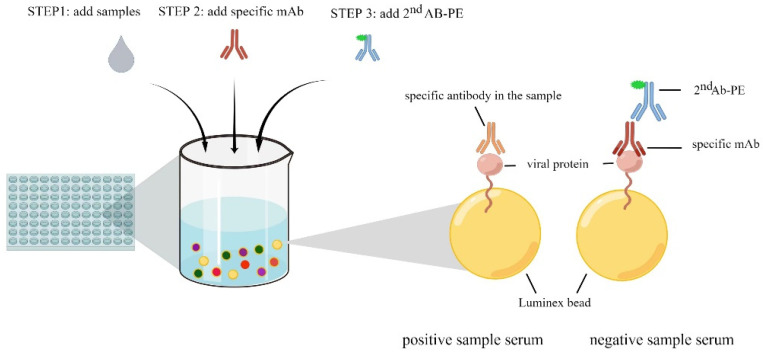
Schematic principle of the Luminex bead-based assay utilized in this study. Appropriate amounts of four recombinant antigens were coated onto each of our Luminex microbead species distinguished by the emitted fluorescence; aliquots of 2000 antigen-coated microbeads/protein were used in assays (step 1). Samples were incubated with mixtures of the four microbead species for 30 min; beads were then washed to remove unbound components (washing steps also followed all further incubation steps) (step 2). Mixtures of four monoclonal antibodies specific for each of the recombinant target proteins were added into each well and incubated for 30 min (step 3). PE-labelled polyclonal antibodies raised in goats against murine IgG were incubated with beads for 30 min. After the final washing, the microbeads were resuspended in assay buffer and left shaking for 15 min. The samples were measured, and data were analyzed in the Luminex-BioPlex 100 instrument.

**Figure 4 pathogens-11-01059-f004:**
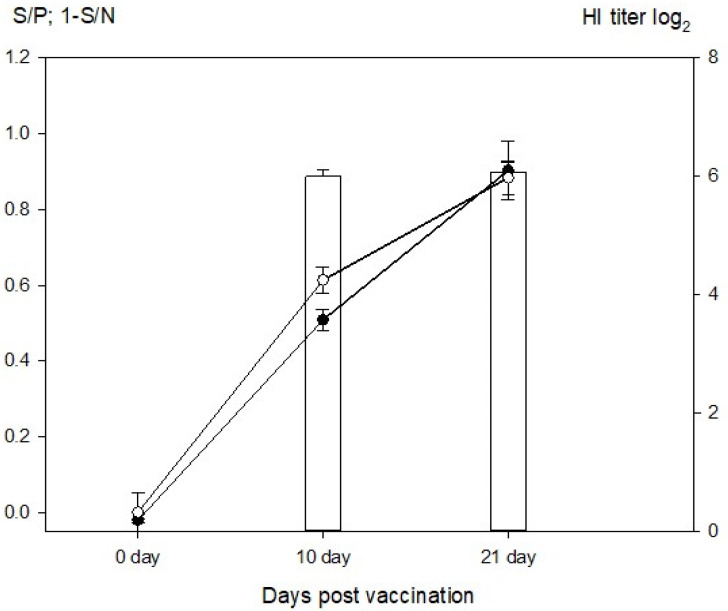
Antibody kinetics in SPF chickens vaccinated against NDV, as measured by HI, commercial ELISA, and 4plex iFMIA assay. Sera were obtained from thirty chickens after immunization with a live-attenuated vaccine based on NDV strain “clone 30” at days 0, 10, and 14 post vaccination. Columns—HI titer against the homologous antigen (NDV clone 30); open circle—commercial ELISA (IDEXX); black dot—4plex iFMIA format.

**Table 1 pathogens-11-01059-t001:** Origin and characteristics of 177 NDV- and AIV-specific sera and 80 SPF sera obtained from chicken and turkeys.

				Flock Immune Status			
Panel	Serum Source	No. of Samples	Species	ND-NP	AI-NP	H5	H7
A	ND vaccination, EXP	70	chicken	+	−	−	−
B	AI H5/ND vaccination, EXP	40	chicken	+	+	+	−
C	AI H7 vaccination, EXP	10	chicken	−	+	−	+
D	AI H7 infection in ND vaccinated birds, EXP, Field	11	turkey	+	+	−	+
E	recND-H7 vaccination, EXP	26	chicken	+	−	−	+
F	AI H9N2 infection, EXP	10	chicken	−	+	−	−
G	AI H9N2 infection in ND vaccinated birds, EXP, Field	10	turkey	+	+	−	−
H	SPF	50/30	chicken/turkey	−	−	−	−

EXP—sera derived from experimental infection/vaccination. Field—sera originating from the field. A—sera collected from specific pathogen-free (SPF) chickens vaccinated with NDV strain “clone 30”. B—sera collected from NDV vaccinated SPF chickens experimentally infected with AIV-H5N2. C—sera collected from SPF chickens vaccinated against AIV H7N7. D—sera collected from NDV vaccinated turkeys (field origin) experimentally infected with AIV-H7N7. E—sera collected from SPF chickens vaccinated with recombinant H7-NDV. F—sera collected from turkeys experimentally infected with AIV H9N2. G—sera collected from NDV vaccinated SPF chickens experimentally infected with AIV H9N2. H—sera collected from SPF chickens and turkeys before experimental infections. (+)—seropositive and (−)—negative on pretests against the indicated antigen.

**Table 2 pathogens-11-01059-t002:** Comparison of performance characteristics (sensitivity, specificity with 95% confidence intervals) of commercial ELISAs and the 4plex iFMIA assay using the hemagglutination inhibition assay (HI) as gold standard.

Target	Assay Format ^a^	Sensitivity	95% CI	Specificity	95% CI	AUC ^b^
NDV Full virion	iELISA (IDEXX)	0.93	0.87–0.96	0.95	0.86–0.99	
NDV-rNP	4plex iFMIA	0.93	0.87–0.96	0.91	0.81–0.97	0.937
AIV Full virion	c-ELISA (iD.Vet)	0.98	0.92–1	0.96	0.92–0.99	
AIV-rNP	4plex iFMIA	0.97	0.91–1	0.96	0.91–0.99	0.987
Subtype H5 virion	c-ELISA (iD.Vet)	0.85	0.69–0.94	0.99	0.97–1	
H5-rHA1	4plex iFMIA	1	0.91–1	0.98	0.95–1	0.996
Subtype H7 virion	c-ELISA (iD.Vet)	0.9	0.76–0.97	0.99	0.97–1	
H7-rHA1	4plex iFMIA	0.95	0.83–1	0.98	0.94–1	0.986

^a^—Sera of groups A–H (Table 1) were used to run the assays. ^b^—AUC values of area under (ROC) curves approaching 1 signal very good performance characteristics based on the ROC curve prediction model [29]. See also Appendix A.

**Table 3 pathogens-11-01059-t003:** Qualitative results of serological investigations (seropositive/total samples analyzed per assay).

Serum Panel ^a^	Detection Antigens										
	NDV NP			AIV NP		HA-H5			HA-H7		
	HI	IDEXX	4Plex iFMIA	iD.Vet	4Plex iFMIA	HI	iD.Vet	4Plex iFMIA	HI	iD.Vet	4Plex iFMIA
ND vaccination chicken	10/10	9/10	9/10	0/10	0/10	0/10	0/10	0/10	0/10	0/10	0/10
AI-H5 vaccination in ND vaccinated chicken	39/40	39/40	33/40	32 (1)/40 ^b^	36/40	40/40	31 (2)/40	40/40	0/40	0/40	0/40
AI H7 vaccination chicken	0/10	0/10	0/10	10/10	10/10	0/10	0/10	0/10	8/10	5 (4)/10	6/10
Recombinant ND-H7 vaccination chicken	26/26	25/26	26/26	0/26	0/26	0/26	0/26	0/26	21/26	14 (5)/26	19/26
Experimental AI-H7 infection in ND vaccinated turkeys	11/11	6/11	3/11	11/11	11/11	0/11	0/11	0/11	10/11	7 (2)/11	9/11
AI H9N2 infection, EXP	0/10	0/10	0/10	10/10	10/10	0/10	0/10	0/10	0/10	0/10	0/10
AI H9N2 infection in ND vaccinated birds, Field	9/10	8/10	9/10	10/10	10/10	0/10	0/10	0/10	0/10	0/10	0/10

^a^—serum details were provided in Table 1. ^b^—brackets indicate questionable results in the commercial ELISA.

**Table 4 pathogens-11-01059-t004:** Performance characteristics of the 4plex iFMIA assay compared to commercial ELISA and HI assay in 250 sera of field origin.

Target	Assay	No. of Positives	Sensitivity ^a^	95% CI	Specificity ^a^	95% CI
NDV	HI	129				
	iELISA (IDEXX)	179	0.83	0.76–0.89	0.82	0.74–0.89
	4plex iFMIA	177	0.96	0.91–0.99	0.56	0.47–0.65
AIV	c-ELISA (iD.Vet)	88	n.a. ^b^		n.a.	
	4plex iFMIA	91	0.87	0.79–0.93	0.95	0.91–0.98
Subtype-H5	HI	0				
	c-ELISA (iD.Vet)	0	No HI-positives ^c^		1	0.98–1
	4plex iFMIA	2	No HI-positives		0.99	0.97–1
Subtype-H7	HI	40				
	c-ELISA (iD.Vet)	40	1	0.91–1	1	0.97–1
	4plex iFMIA	42	1	0.91–1	0.99	0.97–1

^a^—Calculated using the HI results as gold standard. ^b^—Not assessed; no gold standard assay available to control the reactivity of the generic NP ELISA, since positive reactions could have been due to infections with any of the 16 HA subtypes of AIV. ^c^—no H5-specific sera available from the field.

## Data Availability

All relevant data are presented here and in the Appendix A.

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
