# Peer review of "Tetraplex Fluorescent Microbead-Based Immunoassay for the Serodiagnosis of Newcastle Disease Virus and Avian Influenza Viruses in Poultry Sera"

_pathogens, 2022, doi:10.3390/pathogens11091059_

Round 1

Reviewer 1 Report

In the present manuscript (essentially a methodological paper), Zhao and colleagues report the development of a novel tetraplexed fluorescent microbead-based immunoassay (4plex iFMIA) for the simultaneous detection of NDV (NP) or AIV (NP, H5/H7 serotype) specific antibodies in sera from gallinaceous poultry.

There is an urgent need for the development of new versatile, multiplexed assays for the serodiagnosis of H5/H7 AIV infections in domestic poultry, especially in view of the rampant spread of H5 HPAIV over the past few years; thus, the manuscript addresses a timely and import issue.    

The materials and methods are described in sufficient detail, and the authors provide compelling evidence for the sensitivity, specificity and practical use of the multiplexed heterogenous immunoassay presented here (though there is clearly room to improve the assay format and performance).  

Minor comments:

Materials and Methods, Fig. 1: Amino acid lengths of the four different constructs should be indicated in the graphical representation.

Materials and Methods, Table 1: The concomitant use of the letter F for indicating the use of field sera and for designating panel F "AI H9N2 infection, EXP" is somewhat confusing. The use of the various serum panels for evaluating the 4plex iFMIA remains elusive without the information provided in Table S2.    

Supplementary files, Table S2: The table contains important, insightful information and should be included in the manuscript.

Results, Fig. 2: Judging from the antibody reaction pattern and the size of the specific bands, it seems that lanes 2/3 (corresponding to rHA1_H7/H5) have been mixed up.   

Results, Fig. 3: The graphical representation of step b is somewhat confusing and does not reflect the heterogeneity of the tested serum samples (including non-specific antibodies).

The authors should check for consistency in the use of NDV or AMPV when referring to Newcastle disease virus.

Author Response

Dear review,

Thank you for your positive and constructive criticism. 

We have addressed all concerns and suggestion as detailed below. We hope the revised version will find full approval.

Point 1. Materials and Methods, Fig. 1: Amino acid lengths of the four different constructs should be indicated in the graphical representation.

Response 1: Corrected accordingly. Please check Fig. 1.

Point 2. Materials and Methods, Table 1: The concomitant use of the letter F for indicating the use of field sera and for designating panel F "AI H9N2 infection, EXP" is somewhat confusing. The use of the various serum panels for evaluating the 4plex iFMIA remains elusive without the information provided in Table S2.    

Response 2: Instead of letter "F” we have used “Field” to indicate the origin in Table 1 and Table S2 (which is now moved to main text as table 3).

Point 3. Supplementary files, Table S2: The table contains important, insightful information and should be included in the manuscript.

Response 3: We have moved this table to the main text and refer to it as Table 3.

Point 4. Results, Fig. 2: Judging from the antibody reaction pattern and the size of the specific bands, it seems that lanes 2/3 (corresponding to rHA1_H7/H5) have been mixed up. 

 Response 4: Thank you for spotting this error; the labelling has been corrected accordingly.

 Point 5. Results, Fig. 3: The graphical representation of step b is somewhat confusing and does not reflect the heterogeneity of the tested serum samples (including non-specific antibodies).

Response 5: This figure has been revised to provide a better/clearer representation. Pls check the new figure 3.

 Point 6. The authors should check for consistency in the use of NDV or AMPV when referring to Newcastle disease virus.

Response 6: We are now using NDV throughout the manuscript.

Reviewer 2 Report

Dear authors, the study was significant for the advancement of serodetection of avian pathogens.

My general comment is that the manuscript's content was too lengthy, whereby redundancies were detected, and several long sentences were confusing and contained grammar errors. 

My specific comments were as below:

1. Revise the title, i.e. "APMV-1" was not comparable with "Avian Influenza Virus". Suggestion: Tetraplex fluorescent microbead-based immunoassay for the serodetection of avian paramyxovirus-1 and avian influenza virus in poultry sera"

2. Revise the first sentence of the abstract to point up the APMV-1 first, followed by AIV- as represented in the title.

3.  Under materials and methods, the first subtopic (2.1) would be "Ethical approval" whereby you indicate the ethical clearance/approval of the animal handling and experiment. 

4. With regards to point no. 3 above, remove redundant sentences 140-143 and 147-149 on page 4. Maintain the sentence "A total of 257 sera, which included 80 from those SPF control chickens, was used".

5. [Subtopic 2.1, lane 96]- LP change to LPAI, and HP to HPAI.

6. [Subtopic 2.2, lane 11]- TYH: provide the full name and constituents or refer to the product's brand and catalogue number.

7. Figure 1: [H]6 or [H]8?

8. [Subtopic 2.3, lanes 122-123]- Remove "to determine the most appropriate buffer conditions to refold the denatured proteins into an antigenically authentic structure while keeping it solubilized". Please indicate the suitable refolding buffers for the purification of the different ICs under Results.

9. Table 1- Include the total number of samples in figure caption.

10. Table 1- Include indicators for (+) and (-) in figure caption.

11. [Subtopic 2.6, lane 166]- 150+130 did not total up to the number in Table 1. Remove this subtopic as it was considered as irrelevant. The sentence "In Germany, NDV vaccination is compulsory........" could be embedded under Discussion. 

12. [Subtopic 2.10, lanes 212-213]- Revise sentence for grammar.

13. [Subtopic 3.2, lane 300]- indicate the full "MFI" abbreviation.

14. [Subtopic 3.2, lane 307]- typo IAV, change to AIV.

15. Figure 3- Revise and enlarge for better/clearer representation.

16. [Subtopic 3.3, lane 328]- Remove "Receiver operating characteristic".

17. [Subtopic 3.3, lane 331]- Typo 39.4, not 39,4.

18. Discussion, lanes 389-393- Should be under Introduction.

19. Discussion, lanes 394-407 [until ......... AIV-NP antibody test (Table 2, 4)]- Redundant, should be reported under Results. 

20. Discussion, lanes 447-450- Revise sentence to "A further expansion of the current work would be to include possible quantification of viral/antigen loads, and serodetection of antibodies against other AIV subtypes (e.g. the H9), NDV proteins (e.g. the HN) and other significant avian disease viruses (e.g. the infectious bronchitis or infectious bursal disease viruses)".

Author Response

Dear reviewer,

Thank you for your positive and constructive criticism. 

We have addressed all concerns and suggestion as detailed below. We hope the revised version will find full approval.

Point 1. Revise the title, i.e. "APMV-1" was not comparable with "Avian Influenza Virus". Suggestion: Tetraplex fluorescent microbead-based immunoassay for the sero detection of avian paramyxovirus-1 and avian influenza virus in poultry sera"

Response 1: The title has been changed using “Newcastle disease virus” instead and now runs “Tetraplex fluorescent microbead-based immunoassay for the serodetection of Newcastle disease virus and avian influenza viruses in poultry sera”.

Point 2. Revise the first sentence of the abstract to point up the APMV-1 first, followed by AIV- as represented in the title.

Response 2: Amended accordingly.

Point 3.  Under materials and methods, the first subtopic (2.1) would be "Ethical approval" whereby you indicate the ethical clearance/approval of the animal handling and experiment. 

Response 3: Thanks for your suggestion. The ethical permission has been moved to the indicated section accordingly.

Point 4. With regards to point no. 3 above, remove redundant sentences 140-143 and 147-149 on page 4. Maintain the sentence "A total of 257 sera, which included 80 from those SPF control chickens, was used".

Response 4: Corrected as suggested.

Point 5. [Subtopic 2.1, lane 96]- LP change to LPAI, and HP to HPAI.

Response 5: Corrected.

Point 6. [Subtopic 2.2, lane 11]- TYH: provide the full name and constituents or refer to the product's brand and catalogue number.

Response 6: The recipe of TYH medium has been added in section 2.2, lane 121-122: “Induction of expression was achieved in tryptone-yeast-hepes (TYH) medium (20 g tryptone; 10g yeast extract; 5 g NaCl; 1 g MgSO4; 11 g HEPES in 1 liter aq. bidist.) supplemented with IPTG (1.5 mM) and D (+)-biotin (50 µM).”

Point 7. Figure 1: [H]6 or [H]8?

Response 7: It should be [H]8, corrected.

Point 8. [Subtopic 2.3, lanes 122-123]- Remove "to determine the most appropriate buffer conditions to refold the denatured proteins into an antigenically authentic structure while keeping it solubilized". Please indicate the suitable refolding buffers for the purification of the different ICs under Results.

Response 8: The recipe of the optimal refolding buffer was given in lane 135-137. “Finally, an optimized refolding buffer was identified consisting of 50 mM Tris, pH 8.3, 20 mM NaCl, 0.8 mM KCl, 0.8 M L-Arginine, 0.12 M sucrose.”

Point 9. Table 1- Include the total number of samples in figure caption.

Response 9: Caption changed to: “Table 1. Origin and characteristics of 177 NDV- and AIV-specific sera and 80 SPF sera obtained from chicken and turkeys.”

Point 10. Table 1- Include indicators for (+) and (-) in figure caption.

Response 10: Added to the legend: “(+) – seropositive and (-) -negative on pretests against the indicated antigen”

Point 11. [Subtopic 2.6, lane 166]- 150+130 did not total up to the number in Table 1. Remove this subtopic as it was considered as irrelevant. The sentence "In Germany, NDV vaccination is compulsory........" could be embedded under Discussion. 

Response 11: These sera actually were not used for reference/validation purposes but represented true field sera with unknown pretest results. We tried to indicate this by using a new heading for this section as indicated below:

"2.7 Origin of field sera not used for reference purposes

A total of 150 chicken and 130 turkey sera submitted for routine AI or ND serodiagnosis originated from various poultry holdings in Germany and were investigated by 4plex iFMIA assay and other routine assays in this study.”

The indicated sentence has been moved to the discussion.

Point 12. [Subtopic 2.10, lanes 212-213]- Revise sentence for grammar.

Response 12: Corrected. “Next, serum samples diluted 1:20 in 50 µL assay buffer were applied to the plates and incubated for 40 min, then the solution was removed and the beads washed.”

Point 13. [Subtopic 3.2, lane 300]- indicate the full "MFI" abbreviation.

Response 13: The meaning of “MFI” is indicated in section 2.10 (lane 246), “median fluorescence intensity (MFI)”.

Point 14. [Subtopic 3.2, lane 307]- typo IAV, change to AIV.

Response 14: Corrected.

Point 15. Figure 3- Revise and enlarge for better/clearer representation.

Response 15: Revised as suggested.

Point 16. [Subtopic 3.3, lane 328]- Remove "Receiver operating characteristic".

Response 16: Removed.

Point 17. [Subtopic 3.3, lane 331]- Typo 39.4, not 39,4.

Response 17: Corrected.

Point 18. Discussion, lanes 389-393- Should be under Introduction.

Response 18: Has been moved accordingly.

Point 19. Discussion, lanes 394-407 [until ......... AIV-NP antibody test (Table 2, 4)]- Redundant, should be reported under Results. 

Response 19: Removed from the discussion and mentioned in results.

Point 20. Discussion, lanes 447-450- Revise sentence to "A further expansion of the current work would be to include possible quantification of viral/antigen loads, and serodetection of antibodies against other AIV subtypes (e.g., the H9), NDV proteins (e.g., the HN) and other significant avian disease viruses (e.g., the infectious bronchitis or infectious bursal disease viruses)".

Response 20: Revised accordingly.